# Robust Contrastive Cross-modal Hashing with Noisy Labels

Longan Wang
Sichuan University
Chengdu, China
wanglongan@stu.scu.edu.cn

Yang Qin
Sichuan University
Chengdu, China
qinyang.gm@gmail.com

Yuan Sun
Sichuan University
Chengdu, China
sunyuan_work@163.com

Dezhong Peng*
Sichuan University
Chengdu, China
pengdz@scu.edu.cn

Xi Peng
Sichuan University
Chengdu, China
pengx.gm@gmail.com

Peng Hu[†]
Sichuan University
Chengdu, China
penghu.ml@gmail.com

## Abstract

Cross-modal hashing has emerged as a promising technique for retrieving relevant information across distinct media types thanks to its low storage cost and high retrieval efficiency. However, the success of most existing methods heavily relies on large-scale well-annotated datasets, which are costly and scarce in the real world due to ubiquitous labeling noise. To tackle this problem, in this paper, we propose a novel framework, termed Noise Resistance Cross-modal Hashing (NRCH), to learn hashing with noisy labels by overcoming two key challenges, *i.e.*, noise overfitting and error accumulation. Specifically, i) to mitigate the overfitting issue caused by noisy labels, we present a novel Robust Contrastive Hashing loss (RCH) to target homologous pairs instead of noisy positive pairs, thus avoiding overemphasizing noise. In other words, RCH enforces the model focus on more reliable positives instead of unreliable ones constructed by noisy labels, thereby enhancing the robustness of the model against noise; ii) to circumvent error accumulation, a Dynamic Noise Separator (DNS) is proposed to dynamically and accurately separate the clean and noisy samples by adaptively fitting the loss distribution, thus alleviate the adverse influence of noise on iterative training. Finally, we conduct extensive experiments on four widely used benchmarks to demonstrate the robustness of our NRCH against noisy labels for cross-modal retrieval. The code is available at: https://github.com/LonganWANG-cs/NRCH.git.

## CCS Concepts

• **Information systems → Specialized information retrieval**.

## Keywords

Cross-modal Hashing, Noisy Labels, Cross-modal Retrieval

---

*Dezhong Peng is also with Sichuan Newstrong UHD Video Technology Co., Ltd.
[†]Corresponding author

**ACM Reference Format:**
Longan Wang, Yang Qin, Yuan Sun, Dezhong Peng, Xi Peng, and Peng Hu. 2024. Robust Contrastive Cross-modal Hashing with Noisy Labels. In *Proceedings of the 32nd ACM International Conference on Multimedia (MM '24), October 28-November 1, 2024, Melbourne, VIC, Australia.* ACM, New York, NY, USA, 9 pages. https://doi.org/10.1145/3664647.3680564

## 1 Introduction

Cross-modal hashing aims to map different modalities (*e.g.*, image and text) into a common Hamming space, providing an efficient technique for storing and retrieving large-scale databases [15]. The main challenge lies in learning discriminative and compact binary codes by overcoming the heterogeneity gap across distinct modalities. To this end, numerous cross-modal hashing approaches have been proposed and achieved remarkable progress, but they usually implicitly assume that the training labels are free from inaccuracies [44]. However, this assumption is unrealistic in real-world scenarios, where noisy labels are ubiquitous due to various factors, such as data complexity [6], human annotation errors [26], label obfuscation tactics [34], *etc*. Inevitably, the noisy labels could mislead deep hashing models to overfit the noise, degrading retrieval performance.

To address the negative impact of noisy labels, various strategies have been proposed for unimodal learning tasks, such as sample selection [39], loss correction [8], label correction [18], *etc*. Unfortunately, these techniques cannot directly handle the noisy labels in cross-modal learning due to the inherent heterogeneity gap. To address this challenge, some cross-modal methods have been developed to deal with noisy labels by pre-processing [20], robust loss functions [10], early learning regularization [36], *etc*. However, these techniques are tailored for real-value continuous representations, incurring additional computational and storage complexity. Compared with continuous representations, binary codes are more lightweight and efficient, but unreliable labels are more likely to amplify quantization errors, which critically compromises the effectiveness of the hashing model [37]. Thus, it is urgent to study how to conquer the harmful impact of noisy labels to learn discriminative and robust binary codes for cross-modal retrieval.

In order to address this problem, a few methods [32, 37] have been proposed and achieved promising performance. Among them, NrDCMH [32] identifies noise by contrasting feature-label similarity but is limited to noise that only non-class semantic labels are flipped. The latest advancement is CMMQ [37], which favors instances with lower loss values. However, the selection ratio in

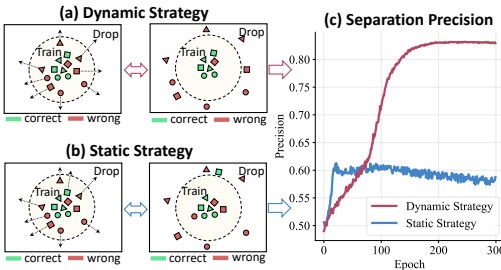

**Figure 1: Key points of noise separation strategies. The dynamic strategy as evidenced in (a) predicts label noise more reliably, enhancing model accuracy and bypassing the decision boundary sensitivity. However, the static strategy prone to error as shown in (b) often excludes valuable samples while keeping noisy ones due to its rigid loss-based filtering. The precision curves of the two strategies are shown in (c).**

this method is fixed and depends on an estimated noise level, which is hard to measure accurately in practice. Moreover, the deep model would exacerbate overfitting due to the over-memorization of noisy labels (see Figure 1), making such a static separation strategy less effective in separating between noisy and clean labels in the later training stages [40]. Therefore, a reliable dynamic label noise separation method is highly needed for robust cross-modal hashing.

To tackle the challenges above, we propose a novel noise-robust cross-modal hashing method called Noise Resistance Cross-modal Hashing (NRCH). The pipeline of our NRCH is illustrated in Figure 2, which consists of two key techniques: Robust Contrastive Hashing loss (RCH) and Dynamic Noise Separator (DNS). More specifically, RCH is proposed to leverage homologous pairs instead of noisy positive pairs and integrate robust contrastive learning with selective negative pairs to mitigate the negative impact of noisy labels. Thus, RCH could learn robust and congruent binary representations across different modalities. On the other hand, DNS is presented to dynamically separate the clean and noisy labels based on the loss distributions. Our DNS uses a Gaussian Mixture model to dynamically identify the dependable samples by evaluating their likelihood of belonging to distributions with lower losses. By combining RCH and DNS, our NRCH could simultaneously distinguish the clean and noisy labels, and learn binary codes robustly for cross-modal retrieval. The main contributions of our NRCH are summarized as follows:

- This paper studies an emerging but less-touched problem in cross-modal hashing, *i.e.*, hashing with noisy labels. To achieve this, we propose a novel Noise Resistance Cross-modal Hashing (NRCH) framework for cross-modal hashing, which ensures robustness from the perspective of loss function and sample dynamic selection.
- A Robust Contrastive Hashing loss (RCH) is proposed to focus on homologous pairs rather than noisy positive ones, embracing more reliable binary representations for cross-modal retrieval.
- We present a novel Dynamic Noise Separator (DNS) to dynamically discriminate the clean and noisy labels based on

the loss distributions, which avoids the expensive manual estimation of noisy levels and mitigates noise overfitting.
- Extensive experiments are conducted on four widely used benchmarks and demonstrate the robustness of the proposed method against noisy labels by comparison with 11 state-of-the-art baselines.

## 2 Related Work

### 2.1 Deep Cross-modal Hashing

Deep Cross-Modal Hashing (CMH) methods [4, 11, 15, 28, 29, 37] aim to project multimodal data into a common Hamming space, providing an efficient solution for the storage and retrieval of large-scale datasets. Although supervised approaches leveraging accurate semantic labels have made remarkable progress, they typically employ losses based on classification, such as cross entropy [37] or similarity-based losses (pairwise [15] and triplet [4] losses). To mitigate the dependency on extensive volumes of well-labeled data, semi-supervised learning strategies [33] have been proposed to utilize a synergy of both labeled and unlabeled data. However, almost all of these methods implicitly assume all modalities are labeled correctly, which is infeasible in real-world scenarios due to ubiquitous labeling noise. To tackle this challenge, some robust methods are presented to learn hashing with noisy labels [32, 37]. For instance, a Noise-robust Deep Cross-modal Hashing method (NrDCMH) is proposed to generate robust binary codes by adjusting data pair weights based on feature and label agreement, whereas it is limited to label noise where only non-class semantic labels are flipped (*i.e.*, additive label noise) [2]. Moreover, Yang et al. utilize confident samples to yield lower losses and optimize training under a fixed forgetting rate [37]. Nonetheless, the process of establishing the appropriate selection ratio depends on manual noise estimation, which remains a challenge in real-world scenarios. Besides, the rigid loss-based filtering still leads to noise overfitting due to its inability to separate noise precisely. Thus, developing dynamic strategies to tackle noisy labels effectively remains an open research area.

### 2.2 Learning with Noisy Labels

Prior studies [6, 10, 26, 37, 40] have deeply delved into knowledge extraction from datasets marred by label noise. A common strategy for enhancing learning efficacy revolves around correcting mislabeled data or loss functions, generally referred to as remedial techniques [8]. Nevertheless, these methods invariably hinge on extensive clean labels to bolster their training structures, which is limited and not cost-effective in real-world applications [26]. To address this issue, various studies have targeted designing strategies that can identify accurately labeled samples directly for training networks, such as MentorNet [39], Co-teaching [9], *etc.* However, these strategies fall short of effectively bridging gaps between diverse modalities, as their design is principally optimized for unimodal tasks. To bridge the diverse modality gaps present in real-world multimedia content, a variety of cross-modal methodologies [10, 20, 23–25, 36] have been put forth. By obtaining a unified real-valued representation for different modalities, these methods can effectively bridge these heterogeneous gaps. Nevertheless, all of these existing methods crafted to manage noisy labels are explicitly tailored for continuous value representations, resulting in added computational

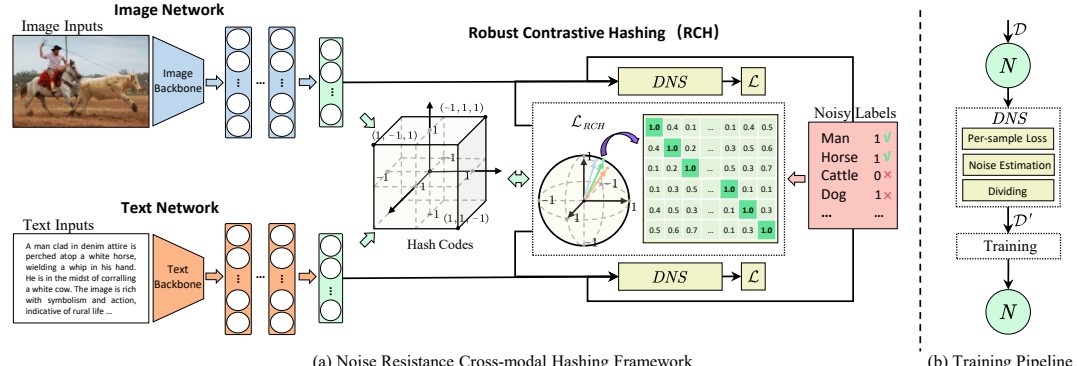

(a) Noise Resistance Cross-modal Hashing Framework                    (b) Training Pipeline

**Figure 2: (a) is the overall framework of our NRCH, which employs a cross-modal network $N = \{f_1, f_2\}$ to learn hashing with noisy labels. (b) is the training pipeline of our method. In NRCH, Robust Contrastive Hashing (RCH) leverages the homologous pairs rather than noisy positive ones and guides $N$ to learn unified hash codes across different modalities through convincing samples selected by Dynamic Noise Separator (DNS). To train the networks $N$ with convincing set $\mathcal{D}'$, DNS discriminates the clean and corrupted labels in $\mathcal{D}$ dynamically by estimating their likelihood to be noise via the designed per-sample loss.**

and storage complexities [37]. Moreover, compact binary representations can easily bring notable quantization errors, which brings a more urgent challenge in cross-modal hashing against noisy labels.

## 3 Method

### 3.1 Problem Formulation

To clarify the presentation, we provide some definitions for cross-modal hashing. In this paper, we take the image and text modalities to study cross-modal hashing with noisy labels. We first represent the multimodal training dataset comprising $N$ examples as $\mathcal{D} = \left\{ \{x_j^i\}_{i=1}^2, y_j \right\}_{j=1}^N$, where $x_j^i$ represents the $j$-th sample from the $i$-th modality, $y_j \in \mathbb{R}^C$ denotes the corresponding noisy label, and $C$ is the number of categories. Note that if the $c$-th element of $y_j$ is 1, it means the $j$-th instance ($\{x_j^i\}_{i=1}^2$) belongs to category $c \in \{1, 2, \cdots, C\}$, otherwise it is 0. For convenience, when above $i = 1$, it denotes that the sample is from the image modality, while $i = 2$ signifies it belongs to the text modality. In addition, we define an indicator of $T_{ij} \in \{0, 1\}$ for the $i$-th image and $j$-th text to show whether they share at least one common category during the training process. $T_{ij} = 1$ means shared, and vice versa.

Given the above $\mathcal{D}$, the objective of cross-modal hashing is to project diverse modalities into a shared Hamming space. Within this space, the unified codes for different modalities are denoted as $\mathcal{B}_i = \left\{ b_j^i \right\}_{j=1}^N$, where $b_j^* \in \{-1, +1\}^L$, $* \in \{1, 2\}$ and $L$ represents the length of hash codes. The Hamming distance is employed to assess the similarity between image and text samples. For any image-text pair $(x_i^1, x_j^2)$, the Hamming distance is defined as $d(b_i^1, b_j^2) = \frac{1}{2} \left( L - \left\langle b_i^1, b_j^2 \right\rangle \right)$, where $\left\langle b_i^1, b_j^2 \right\rangle$ means the inner product. Obviously, when the image and the text exhibit similarity in semantics with $T_{ij} = 1$, the Hamming distance should be small. Conversely, if $T_{ij} = 0$, the Hamming distance should be large.

To learn unified binary representations, we employ distinct hash functions tailored for diverse modalities. More specifically, the hash functions take the form of $f_1(\cdot, \Theta_1)$ and $f_2(\cdot, \Theta_2)$ for image

and text modalities, respectively, where $\Theta_1$ and $\Theta_2$ denote the corresponding modality-specific network parameters. For brevity, $f_*(\cdot, \Theta_*), * \in \{1, 2\}$ is denoted as $f_*$ or $f_*(\cdot)$ in the following. In our NRCH framework, for $i$-th instance, the outcomes of these hash functions are denoted as $h_i^1 = f_1(x_i^1)$ and $h_i^2 = f_2(x_i^2)$. The corresponding binary representation of a sample is derived by applying the sign function [15] to $h_i^*$:

$$b_i^* = \text{sign}\left(h_i^*\right), * \in \{1, 2\}. \tag{1}$$

### 3.2 Robust Contrastive Hashing

To enhance the efficiency of cross-modal hashing, it is imperative to promote the proximity of similar samples and the separation of dissimilar samples from different modalities. Driven to achieve this goal and informed by the proven success of the triplet loss as evidenced in [4, 16]. Given a mini-batch $\mathcal{D}_n \subseteq \mathcal{D}$, we have:

$$\mathcal{L}_r^*(\mathcal{D}_n) = \frac{1}{n^3} \sum_{i=1}^n \sum_{k=1}^n \sum_{j=1}^n T_{ik}\left(1 - T_{ij}\right) \max\left(0, m + S_{ij}^* - S_{ik}^*\right), \tag{2}$$

where, $* \in \{12, 21\}$, $m$ is a positive margin value, and $n \ll N$ is the size of a mini-batch, $i.e.$, $|\mathcal{D}_n|$. Meanwhile, to address false pairs, we employ the similarity matrix of $S^*$ proposed in [11], as follows:

$$S_{ij}^* = \begin{cases} \mathcal{M}_{ij}^*, & \mathcal{M}_{ii}^* - \mathcal{M}_{ij}^* \leqslant m \\ \mathcal{M}_{ij}^* - \xi, & \text{otherwise} \end{cases}, \tag{3}$$

where $* \in \{12, 21\}$, $\mathcal{M}_{ij}^{12} = \left\langle h_i^1, h_j^2 \right\rangle$, $\mathcal{M}_{ij}^{21} = \left\langle h_i^2, h_j^1 \right\rangle$, and $m$ is a positive margin value. Obviously, Equation (3) capitalizes on the consistent similarity of identical samples across distinct modalities, as reflected diagonally, to reduce the effects of unreliable pairs elsewhere in the similarity matrix. This strategy mitigates the negative impact of false pairs caused by the noisy labels [11] since considering all negative pairs within a soft margin. Thus, we have:

$$\begin{aligned} &\sum_{j=1}^n (1 - T_{ij}) \max\left(0, m + S_{ij}^* - S_{ik}^*\right) \\ &\leq n \left( m + \log \left( \sum_{j=1, j \neq k}^n e^{S_{ij}^*(1 - T_{ij})} + e^{[S_{ik}^*]_+} \right) - S_{ik}^* \right), \end{aligned} \tag{4}$$

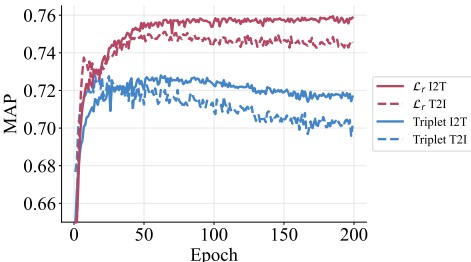

**Figure 3: MAP scores versus epochs for Triplet loss and $\mathcal{L}_r$ on the MIRFlickr-25K dataset with 50% noise and 64 bit code.**

where $* \in \{12, 21\}$. The demonstration procedure is as follows:

$$
\begin{aligned}
& \sum_{j=1}^{n} \left(1 - T_{ij}\right) \max\left(0, m + S_{ij}^* - S_{ik}^*\right) \\
& = |\mathcal{A}_{ik}| \, m + \sum_{S_{ij}^* \in \mathcal{A}_{ik}} S_{ij}^* \left(1 - T_{ij}\right) - |\mathcal{A}_{ik}| \, S_{ik}^* \\
& \leq |\mathcal{A}_i| \left(m + \max_{j=1,\ldots,n} \left(S_{ij}^* \left(1 - T_{ij}\right)\right) - S_{ik}^*\right) \\
& = |\mathcal{A}_i| \left(m + \max_{j=1,\ldots,n} \left(\log\left(e^{S_{ij}^* (1 - T_{ij})}\right)\right) - S_{ik}^*\right) \\
& \leq |\mathcal{A}_i| \left(m + \log\left(\sum_{j=1}^{n} e^{S_{ij}^* (1 - T_{ij})}\right) - S_{ik}^*\right) \\
& \leq n \left(m + \log\left(\sum_{j=1, j \neq k}^{n} e^{S_{ij}^* (1 - T_{ij})} + e^{[S_{ik}^*]_+}\right) - S_{ik}^*\right) \\
& = n \left(m + \log\left(\sum_{H_{ij}^* \in \mathcal{H}_{ik}^*} e^{H_{ij}^*}\right) - S_{ik}^*\right),
\end{aligned}
\tag{5}
$$

where $\mathcal{H}_{ik}^* = \{S_{ij}^* | T_{ij} = 0, j \neq k, j = 1, 2, \cdots, n\} \cup \{[S_{ij}^*]_+ | j = k\}$, $\mathcal{A}_i = \left\{S_{ij}^* \mid S_{ik}^* - S_{ij}^* \leq m; i \neq k, j = 1, 2, \ldots, n; T_{ij} = 0\right\}$, and $|\mathcal{A}_i|$ denotes the length of set $\mathcal{A}_i$. Building upon Equation (4), we can deduce the following inequalities:

$$
\mathcal{L}_r^*(\mathcal{D}_n) \leq \frac{1}{n^2} \sum_{i=1}^{n} \sum_{k=1}^{n} T_{ik}(m + \log(\sum_{H_{ij}^* \in \mathcal{H}_{ik}^*} e^{H_{ij}^*}) - S_{ik}^*), \tag{6}
$$

where $* \in \{12, 21\}$. Consequently, based on Equation (6), we can reformulate the minimization optimization of Equation (2) by the loss function as follows:

$$
\begin{aligned}
\mathcal{L}_r(\mathcal{D}_n) = & \frac{1}{n^2} \sum_{i=1}^{n} \sum_{k=1}^{n} T_{ik}(m + \log(\sum_{H_{ij}^{12} \in \mathcal{H}_{ik}^{12}} e^{H_{ij}^{12}}) - S_{ik}^{12}) \\
& + \frac{1}{n^2} \sum_{i=1}^{n} \sum_{k=1}^{n} T_{ik}(m + \log(\sum_{H_{ij}^{21} \in \mathcal{H}_{ik}^{21}} e^{H_{ij}^{21}}) - S_{ik}^{21}).
\end{aligned}
\tag{7}
$$

To streamline the computations, we only consider diagonals as:

$$
\begin{aligned}
\mathcal{L}_r(\mathcal{D}_n) = & \frac{1}{n} \sum_{i=1}^{n} (m + \log(\sum_{H_{ij}^{12} \in \mathcal{H}_i^{12}} e^{H_{ij}^{12}}) - S_{ii}^{12}) \\
& + \frac{1}{n} \sum_{i=1}^{n} (m + \log(\sum_{H_{ij}^{21} \in \mathcal{H}_i^{21}} e^{H_{ij}^{21}}) - S_{ii}^{21}),
\end{aligned}
\tag{8}
$$

where $\mathcal{H}_i^* = \{S_{ij}^* | T_{ij} = 0, j \neq i, j = 1, 2, \cdots, n\} \cup \{[S_{ij}^*]_+ | j = i\}$ and $* \in \{12, 21\}$. As demonstrated in Figure 3, unlike the Triplet loss where significant performance degradation typically indicates overfitting, $\mathcal{L}_r$ exhibits stronger robustness by maintaining stable MAP

scores through the training process. Finally, our Robust Contrastive Hashing loss (RCH) is as follows:

$$
\mathcal{L}_{RCH}(\mathcal{D}_n) = \lambda \mathcal{L}_r(\mathcal{D}_n) + (1 - \lambda)R(\mathcal{D}_n), \tag{9}
$$

where $R(\mathcal{D}_n)$ is a binary regularization term. $R(\mathcal{D}_n)$ is designed to diminish the quantization error of the acquired binary codes, and $\lambda \in (0, 1)$ stands as a hyperparameter governing the contributions. To forestall overfitting in the subsequent training, we impose restrictions on $|\boldsymbol{b}_j^i| = 1$, and the comprehensive $R(\mathcal{D}_n)$ is defined as:

$$
R(\mathcal{D}_n) = \frac{1}{2 \cdot n \cdot L} \sum_{j=1}^{n} \sum_{k=1}^{L} \sum_{i=1}^{2} \left(|b_{jk}^i| - \frac{1}{L}\right), \tag{10}
$$

where $b_{jk}^i$ is the $k$-th element of $\boldsymbol{b}_j^i$.

## 3.3 Dynamic Noise Separator

Although RCH provides a more stable and robust loss mode, it lacks explicitly separating the noisy labels during training, which can still cause noise overfitting, thus leading to performance degradation (See Section 4.6). To train with convincing samples, recent work [37] focused on a static strategy by discarding unreliable samples at a fixed ratio. However, configuring such a precise forgetting ratio necessitates manual estimation of noise levels, which is challenging and unreliable in real-world applications. Besides, it is easy to cause error accumulations. Inspired by the memorization effect [1, 40] of deep neural networks (DNNs), *i.e.*, the loss values for clean samples are commonly lower than that of noisy ones at the early stage, which can be used to identify the instances with corrupt labels dynamically. Given that, we design an effective label noise separator termed Dynamic Noise Separator (DNS) for robust cross-modal hashing. More specifically, DNS leverages the contrast in loss distribution between clean and noisy samples per instance to gauge its likelihood of being corrupted and dynamically select confident samples to train our network.

To calculate the discriminative per-sample loss for selection, we first exploit a learnable parameter matrix $\mathbf{W} \in \mathbb{R}^{C \times L}$ to encode all categories into corresponding binary representations $\{\hat{\boldsymbol{b}}_k\}_{k=1}^{C}$, where $\hat{\boldsymbol{b}}_k \in \{-1, +1\}^L$ and $\hat{\boldsymbol{b}}_k = \text{sign}(\mathbf{W}_{[k,:]})$. Thus, we can connect the binary representations between the sample and each category to formulate per-sample loss $\ell = \{\ell_j\}_{j=1}^{N}$ for $N$ training instance as:

$$
\ell_j = \max\left(\left\{\sum_{i=1}^{2} y_{jk} \left(\tilde{y}_{jk} - \left\langle \boldsymbol{b}_j^i, \hat{\boldsymbol{b}}_k \right\rangle\right)^2\right\}_{k=1}^{C}\right), \tag{11}
$$

where $\boldsymbol{b}_j^i$ is the binary representation of the $j$-th sample from the $i$-th modality, $y_{jk} \in \{0, 1\}$ is the $k$-th element of $\boldsymbol{y}_j$, and $\tilde{y}_{jk} = 2y_{jk} - 1 \in \{-1, 1\}$ that aligns binary representations. Furthermore, the operation of $\max(\cdot)$ is to tackle noise within multi-label samples by selecting the highest loss value as an indicator of potential label corruption.

Then, the per-sample loss is put into a two-component Gaussian Mixture Model (GMM) [12, 22] to separate instances with noisy labels by fitting the loss distributions for the entire dataset. The GMM is defined as follows:

$$
p(\ell \mid \theta) = \sum_{k=1}^{K} \beta_k \phi(\ell \mid k), \tag{12}
$$

where $\beta_k$ denotes the mixture weight and $\phi(\ell \mid k)$ encapsulates the probability density for the $k$-th Gaussian component in the model. Drawing upon the DNNs' tendency to memorize cleaner data [40] at the early training, we assign the component with a lower mean value as a clean subset and the other as a noisy subset, respectively. Following [18], we employ the Expectation-Maximization strategy to optimize the two-component GMM. Finally, we calculate the posterior probability to represent the likelihood of the $j$-th instance being noise-free, with $k$ symbolizing the Gaussian component exhibiting the lower mean. The formulation is defined as:

$$w_j = p(k \mid \ell_j) = \frac{p(k) \cdot p(\ell_j \mid k)}{p(\ell_j)}. \tag{13}$$

To separate samples with noisy labels, we employ a threshold on $\mathcal{W} = \{w_j\}_{j=1}^N$ to divide the entire training dataset into clean and noisy subsets. Unlike using the fixed threshold empirically [18], we recommend a threshold that gradually increases with the number of training epochs to obtain the optimal selection for convincing clean data based on the DNNs fitting rule. The convincing samples for training can be selected from the training set $\mathcal{D}$ as follows:

$$\mathcal{D}' = \left\{ \begin{array}{c} (x_i^1, x_i^2, y_i, w_i) \mid w_i \geq \text{Threshold}(t), \\ \forall (x_i^1, x_i^2, y_i, w_i) \in (\mathcal{D}, \mathcal{W}) \end{array} \right\}, \tag{14}$$

where $\mathcal{D}$ denotes the training dataset, $t$ is the current training epoch, and $\text{Threshold}(t)$ is the function to set the dynamic filler threshold.

As discussed in [37], DNNs are capable of capturing straightforward and clean patterns amidst noisy labels at first, yet tend to overfit as training epochs increase. Thus, DNS is initiated with a lower threshold to include a broad array of examples at the start. Then, to ensure the retention of clean samples while filtering out the noisy ones before networks memorize them, $\text{Threshold}(t)$ is gradually heightened to $\xi \in (0, 1)$, which are delineated as follows:

$$\text{Threshold}(t) = \min(\frac{t}{T_k}\xi, \xi). \tag{15}$$

### 3.4 Optimization

Thanks to our RCH and DNS, our NRCH can improve the robustness of the loss function while dynamically selecting convincing samples for robust cross-modal hash training. Before dynamic selection, we first conduct a warmup process on $\mathcal{D}$ to reach initial convergence. Given a mini-batch $\mathcal{D}_n \subseteq \mathcal{D}$, $\mathcal{L}_{warm}$ to warmup is defined as:

$$\mathcal{L}_{warm} = \mathcal{L}_{RCH}(\mathcal{D}_n) + \mathcal{L}_W(\mathcal{D}_n), \tag{16}$$

where $\mathcal{L}_D(\mathcal{D}_n)$ is the RCH loss as shown in Equation (9) and $\mathcal{L}_W(\mathcal{D}_n)$ is the loss to optimize $\mathbf{W}$. $\mathcal{L}_W(\mathcal{D}_n)$ is defined as:

$$\mathcal{L}_W(\mathcal{D}_n) = \frac{1}{2 \cdot n \cdot C} \sum_{(X_j, y_j) \in \mathcal{D}_n} \sum_{k=1}^C \sum_{i=1}^2 \left( \tilde{y}_{jk}^i - \left\langle b_j^i, \hat{b}_k \right\rangle \right)^2, \tag{17}$$

where $X_j = \{x_j^i\}_{i=1}^2$. After achieving the initial convergence by $\mathcal{L}_{warm}$, we exploit DNS to select the convincing samples as shown in Equation (14) and then use them to train the model robustly. Given a mini-batch $\mathcal{D}_n \subseteq \mathcal{D}$, the final loss is defined as:

$$\mathcal{L}_{final}(\mathcal{D}_n') = \mathcal{L}_{RCH}(\mathcal{D}_n') + \mathcal{L}_W(\mathcal{D}_n'), \tag{18}$$

where $\mathcal{D}_n' = \mathcal{D}_n \cap \mathcal{D}'$. The training process of NRCH is shown in Algorithm 1.

---

**Algorithm 1** Noise Resistance Cross-modal Hashing

**Require:** The noisy training set $\mathcal{D}$, the code length $L$, the network $N = \{f_1(\cdot, \Theta_1), f_2(\cdot, \Theta_2)\}$, the learnable matrix $\mathbf{W}$, the maximal epoch number $T_{\max}$, and the warmup epoch number $T_{warm}$;

1: Randomly initialize network parameters $\{\Theta_i\}_{i=1}^2$ and $\mathbf{W}$;
2: **for** $epoch = 1$ to $T_{\max}$ **do**
3:    **if** $epoch > T_{warm}$ **then**
4:       Select the conniving subset $\mathcal{D}'$ from $\mathcal{D}$ by DNS;
5:    **end if**
6:    **for** $\mathcal{D}_n$ in mini-batches sampled from $\mathcal{D}$ **do**
7:       **if** $epoch \leq T_{warm}$ **then**
8:          Compute $\mathcal{L}_{warm}(\mathcal{D}_n)$ by Equation (16);
9:          Optimize parameters and $\mathbf{W}$ through backpropagation;
10:       **else**
11:          Select the convincing data by $\mathcal{D}_n' = \mathcal{D}_n \cap \mathcal{D}'$;
12:          Compute $\mathcal{L}_{final}(\mathcal{D}_n')$ by Equation (18);
13:          Optimize parameters and $\mathbf{W}$ through backpropagation;
14:       **end if**
15:    **end for**
16: **end for**

**Ensure:** Network parameters $\{\Theta_i\}_{i=1}^2$ and $\mathbf{W}$;

---

## 4 Experiments

To evaluate our NRCH framework, we conduct extensive experiments on four widely-used benchmark datasets, *i.e.*, MIRFlickr-25K [13], IAPR TC-12 [5], NUS-WIDE [3], and MS-COCO [19].

### 4.1 Datasets

In this section, we mainly introduce the used benchmark cross-modal datasets for experiments. The details are as follows:

**MIRFlickr-25K** [13] comprises 25,000 paired instances, each a duo of an image and its associated textual tags, classified into 24 distinct semantic categories with multi-label annotations. Post-removal of instances sans classification details, the dataset distilled to 20,015 pairs for our experiments.

**IAPR TC-12** [5] is a repository of 20,000 image-text pairs, each labeled with 255 distinct semantic categories in a multi-label format. Distinctively, our experiments utilize the complete dataset.

**NUS-WIDE** [3] encompasses 269,648 images, each annotated across 255 multi-label semantic categories. For our investigative work, we have selectively harvested 200,421 image-text pairs that represent the 21 most prevalent categories.

**MS-COCO** [19] is a collection of 123,287 images, each accompanied by five descriptive sentences, and organized into 80 distinct categories. Following the exclusion of pairs lacking labels, our experimental dataset comprises 122,218 image-text pairs.

For **MIRFLICKR-25K** and **IAPR TC-12**, we set aside 2,000 data points randomly as the test (query) dataset, with the residual serving as the retrieval (database) dataset, from which we further distill a training subset of 10,000 points. In the case of **NUS-WIDE**, our test dataset consists of 2,100 points, with 5,000 segregated for training from the retrieval dataset. For **MS-COCO**, 5,000 points are sampled for testing and 10,000 are reserved for training purposes.

**Table 1: The performance comparison in terms of average MAP scores of T2I and I2T tasks on the NUS-WIDE and MS-COCO datasets. The highest and the second-highest scores are in bold and underlined, respectively.**

| Dataset | Method | 20% | | | | 50% | | | | 80% | | | |
|---|---|---|---|---|---|---|---|---|---|---|---|---|---|
| | | 16bit | 32bit | 64bit | 128bit | 16bit | 32bit | 64bit | 128bit | 16bit | 32bit | 64bit | 128bit |
| NUS-WIDE | DJSRH (ICCV'19) | 0.418 | 0.458 | 0.476 | 0.506 | 0.418 | 0.458 | 0.476 | 0.506 | 0.418 | 0.458 | 0.476 | 0.506 |
| | DGCPN (AAAI'21) | 0.575 | 0.597 | 0.624 | 0.634 | 0.575 | 0.597 | 0.624 | 0.634 | 0.575 | 0.597 | 0.624 | 0.634 |
| | UCCH (TAPMI'23) | 0.576 | 0.598 | 0.625 | 0.637 | 0.576 | 0.598 | 0.625 | 0.637 | 0.576 | 0.598 | 0.625 | 0.637 |
| | DCMH (CVPR'17) | 0.492 | 0.488 | 0.482 | 0.455 | 0.422 | 0.424 | 0.426 | 0.425 | 0.400 | 0.416 | 0.420 | 0.415 |
| | ADAH (ECCV'18) | 0.524 | 0.491 | 0.518 | 0.532 | 0.464 | 0.481 | 0.476 | 0.477 | 0.434 | 0.449 | 0.455 | 0.461 |
| | CPAH (TIP'20) | 0.556 | 0.569 | 0.560 | 0.579 | 0.478 | 0.496 | 0.512 | 0.512 | 0.469 | 0.469 | 0.463 | 0.470 |
| | PIP (SIGIR'21) | 0.556 | 0.603 | 0.599 | 0.597 | 0.566 | 0.594 | 0.591 | 0.594 | 0.575 | 0.596 | 0.595 | 0.602 |
| | CMMQ (CVPR'22) | 0.633 | 0.637 | 0.646 | 0.655 | 0.583 | 0.600 | 0.609 | 0.613 | 0.547 | 0.582 | 0.599 | 0.612 |
| | DCHUC (TKDE'22) | 0.603 | 0.601 | 0.588 | 0.580 | 0.596 | 0.587 | 0.589 | 0.581 | 0.575 | 0.589 | 0.589 | 0.583 |
| | MIAN (TKDE'23) | 0.590 | 0.599 | 0.603 | 0.607 | 0.445 | 0.457 | 0.456 | 0.452 | 0.391 | 0.396 | 0.404 | 0.395 |
| | LtCMH (AAAI'23) | 0.492 | 0.506 | 0.544 | 0.567 | 0.472 | 0.500 | 0.538 | 0.554 | 0.507 | 0.537 | 0.554 | 0.565 |
| | **Our NRCH** | **0.658** | **0.679** | **0.683** | **0.685** | **0.639** | **0.657** | **0.667** | **0.677** | **0.605** | **0.611** | **0.627** | **0.641** |
| MS-COCO | DJSRH (ICCV'19) | 0.485 | 0.527 | 0.553 | 0.579 | 0.485 | 0.527 | 0.553 | 0.579 | 0.485 | 0.527 | 0.553 | 0.579 |
| | DGCPN (AAAI'21) | 0.591 | 0.613 | 0.623 | 0.631 | 0.591 | 0.613 | 0.623 | 0.631 | 0.591 | 0.613 | 0.623 | 0.631 |
| | UCCH (TAPMI'23) | 0.569 | 0.581 | 0.594 | 0.623 | 0.569 | 0.581 | 0.594 | 0.623 | 0.569 | 0.581 | 0.594 | 0.623 |
| | DCMH (CVPR'17) | 0.544 | 0.581 | 0.585 | 0.599 | 0.478 | 0.472 | 0.475 | 0.458 | 0.394 | 0.385 | 0.357 | 0.351 |
| | ADAH (ECCV'18) | 0.464 | 0.478 | 0.477 | 0.483 | 0.469 | 0.470 | 0.473 | 0.486 | 0.463 | 0.472 | 0.473 | 0.478 |
| | CPAH (TIP'20) | 0.547 | 0.598 | 0.605 | 0.613 | 0.543 | 0.550 | 0.556 | 0.573 | 0.515 | 0.518 | 0.516 | 0.517 |
| | PIP (SIGIR'21) | 0.538 | 0.575 | 0.588 | 0.597 | 0.513 | 0.554 | 0.599 | 0.603 | 0.500 | 0.522 | 0.565 | 0.591 |
| | CMMQ (CVPR'22) | 0.613 | 0.639 | 0.641 | 0.641 | 0.595 | 0.623 | 0.625 | 0.632 | 0.601 | 0.618 | 0.625 | 0.635 |
| | DCHUC (TKDE'22) | 0.558 | 0.493 | 0.552 | 0.484 | 0.552 | 0.497 | 0.488 | 0.486 | 0.550 | 0.485 | 0.491 | 0.556 |
| | MIAN (TKDE'23) | 0.571 | 0.573 | 0.603 | 0.587 | 0.498 | 0.499 | 0.523 | 0.545 | 0.445 | 0.459 | 0.479 | 0.470 |
| | LtCMH (AAAI'23) | 0.553 | 0.589 | 0.615 | 0.629 | 0.547 | 0.572 | 0.610 | 0.627 | 0.559 | 0.597 | 0.609 | 0.615 |
| | **Our NRCH** | **0.637** | **0.649** | **0.669** | **0.673** | **0.647** | **0.663** | **0.681** | **0.686** | **0.646** | **0.658** | **0.675** | **0.690** |

## 4.2 Implementation Details

In our NRCH, we utilize the pre-trained VGG19 [14] on ImageNet as the convolutional neural network (CNN) backbone for processing images. Meanwhile, we utilize the pre-trained Doc2Vec [17] model as the backbone for processing textual data. To jointly learn shared representations across modalities, three hidden layers are stacked on the backbone for image modality, while two for text modality. Each fully collected (FC) layer is followed by a Rectified Linear Unit (ReLU) layer, except for the last layer. Within these FC structures, a consistent count of 8,192 units is maintained in the hidden layers, culminating in an output layer calibrated to $L$, which symbolizes the dimensional scale of the shared space. Subsequently, the RMSprop algorithm [30] is utilized as the training optimizer for our NRCH model. We standardize the maximum number of epochs $T_{\max}$ at 100 and warmup epochs $T_{warm}$ at 20 for every dataset, respectively, with the hyperparameter $\lambda$ established at 0.6 and margin $m$ at 0.2. We determine the initial rate of learning $\eta$ to be $1e-5$ and the batch size $n$ to 128. $T_k$ is uniformly assigned a value of 100 and the threshold $\xi$ is maintained at 0.3. To ensure a fair comparison with baselines, all backbones remain frozen during the training stage. Our NRCH is implemented using the PyTorch framework [21] and trained on a single RTX3090 24GB GPU.

## 4.3 Experimental Setup

To evaluate the performance, we report the results of two cross-modal retrieval tasks, i.e., image-to-text retrieval (I2T) task and text-to-image retrieval (T2I) task. More specifically, the I2T task is to fetch relevant text by a given image query based on the hamming distance. Conversely, the I2T task retrieves the relevant image using a text query. Like [11, 31], we use the widely used Mean Average Precision (MAP) as the assessment standard to evaluate the retrieval performance. The MAP score is the average value of Average Precision (AP) scores for each query, which is widely acknowledged for evaluating retrieval since it concurrently takes into account both the precision and ranking of yielded outcomes. It's noteworthy that we calculate MAP scores throughout all retrieval outcomes in trials with bit lengths configured to 16, 32, 64, and 128. Moreover, to thoroughly assess the robustness of the approaches, we introduce mixed symmetric label noise [2], with noise rates established at 20%, 50%, and 80% in our experiments.

## 4.4 Comparison with State-of-the-Arts

To demonstrate the superiority and robustness of our method, we compare the proposed NRCH with 11 state-of-the-art methods on four widely used datasets, including the unsupervised methods: DJSRH [27], DGCPN [38], and UCCH [11]; the supervised methods: DCMH [15], ADAH [42], CPAH [35], PIP [41], CMMQ [37], DCHUC [31], MIAN [43] and LtCMH [7]. Significantly, the CMMQ method is specifically proposed to address challenges posed by noisy labels. The average MAP scores for the I2T and T2I tasks are presented in Tables 1 and 2. From the results in these tables, it becomes clear that our NRCH surpasses all baselines consistently across the entirety of the four datasets. Besides, the Figure 4 illustrating precision-recall curves is plotted on code lengths of 64 bits amid a 50% noise rate, both of which additionally elucidate the effectiveness and robustness of our NRCH on T2I and I2T tasks. Through evaluating the region under the precision-recall curves, it becomes apparent that our NRCH consistently surpasses all alternative state-of-the-art approaches in both I2T and T2I tasks.

**Table 2: The performance comparison in terms of average MAP scores of T2I and I2T tasks on the IAPR TC-12 and MIRFlickr-25K datasets. The highest and the second-highest scores are in bold and underlined, respectively.**

| Dataset | Method | 20% | | | | 50% | | | | 80% | | | |
|---|---|---|---|---|---|---|---|---|---|---|---|---|---|
| | | 16bit | 32bit | 64bit | 128bit | 16bit | 32bit | 64bit | 128bit | 16bit | 32bit | 64bit | 128bit |
| IAPR TC-12 | DJSRH (ICCV'19) | 0.368 | 0.397 | 0.421 | 0.434 | 0.368 | 0.397 | 0.421 | 0.434 | 0.368 | 0.397 | 0.421 | 0.434 |
| | DGCPN (AAAI'21) | 0.421 | 0.448 | 0.464 | 0.467 | 0.421 | 0.448 | 0.464 | 0.467 | 0.421 | 0.448 | 0.464 | 0.467 |
| | UCCH (TAPMI'23) | 0.418 | 0.465 | 0.465 | 0.468 | 0.418 | 0.465 | 0.465 | 0.468 | 0.418 | 0.465 | 0.465 | 0.468 |
| | DCMH (CVPR'17) | 0.424 | 0.428 | 0.416 | 0.416 | 0.414 | 0.411 | 0.404 | 0.394 | 0.369 | 0.370 | 0.365 | 0.359 |
| | ADAH (ECCV'18) | 0.421 | 0.432 | 0.449 | 0.448 | 0.408 | 0.417 | 0.443 | 0.444 | 0.414 | 0.410 | 0.429 | 0.433 |
| | CPAH (TIP'20) | 0.450 | 0.466 | 0.466 | 0.473 | 0.441 | 0.453 | 0.457 | 0.462 | 0.422 | 0.449 | 0.456 | 0.458 |
| | PIP (SIGIR'21) | 0.439 | 0.452 | 0.463 | 0.479 | 0.414 | 0.453 | 0.466 | 0.477 | 0.426 | 0.448 | 0.461 | 0.473 |
| | CMMQ (CVPR'22) | 0.418 | 0.446 | 0.467 | 0.469 | 0.413 | 0.445 | 0.462 | 0.471 | 0.424 | 0.438 | 0.456 | 0.460 |
| | DCHUC (TKDE'22) | 0.449 | 0.451 | 0.449 | 0.448 | 0.447 | 0.450 | 0.447 | 0.448 | 0.424 | 0.439 | 0.451 | 0.447 |
| | MIAN (TKDE'23) | 0.440 | 0.445 | 0.455 | 0.435 | 0.422 | 0.429 | 0.437 | 0.445 | 0.403 | 0.419 | 0.428 | 0.431 |
| | LtCMH (AAAI'23) | 0.420 | 0.435 | 0.449 | 0.457 | 0.419 | 0.436 | 0.448 | 0.457 | 0.414 | 0.439 | 0.444 | 0.455 |
| | **Our NRCH** | **0.498** | **0.527** | **0.546** | **0.552** | **0.494** | **0.526** | **0.542** | **0.547** | **0.488** | **0.518** | **0.534** | **0.544** |
| MIRFlickr-25K | DJSRH (ICCV'19) | 0.608 | 0.619 | 0.637 | 0.645 | 0.608 | 0.619 | 0.637 | 0.645 | 0.608 | 0.619 | 0.637 | 0.645 |
| | DGCPN (AAAI'21) | 0.691 | 0.694 | 0.708 | 0.718 | 0.691 | 0.694 | 0.708 | 0.718 | 0.691 | 0.694 | 0.708 | 0.718 |
| | UCCH (TAPMI'23) | 0.690 | 0.712 | 0.715 | 0.718 | 0.690 | 0.712 | 0.715 | 0.718 | 0.690 | 0.712 | 0.715 | 0.718 |
| | DCMH (CVPR'17) | 0.703 | 0.701 | 0.703 | 0.698 | 0.651 | 0.645 | 0.639 | 0.630 | 0.629 | 0.622 | 0.617 | 0.616 |
| | ADAH (ECCV'18) | 0.724 | 0.727 | 0.736 | 0.733 | 0.706 | 0.712 | 0.718 | 0.712 | 0.602 | 0.614 | 0.606 | 0.607 |
| | CPAH (TIP'20) | 0.704 | 0.702 | 0.702 | 0.702 | 0.671 | 0.673 | 0.678 | 0.665 | 0.633 | 0.669 | 0.657 | 0.647 |
| | PIP (SIGIR'21) | 0.683 | 0.692 | 0.694 | 0.704 | 0.663 | 0.697 | 0.696 | 0.702 | 0.685 | 0.682 | 0.702 | 0.702 |
| | CMMQ (CVPR'22) | 0.726 | 0.730 | 0.736 | 0.738 | 0.698 | 0.718 | 0.720 | 0.724 | 0.694 | 0.711 | 0.716 | 0.719 |
| | DCHUC (TKDE'22) | 0.740 | 0.736 | 0.738 | 0.734 | 0.734 | 0.740 | 0.734 | 0.730 | 0.723 | 0.721 | 0.734 | 0.732 |
| | MIAN (TKDE'23) | 0.738 | 0.741 | 0.750 | 0.753 | 0.677 | 0.689 | 0.690 | 0.693 | 0.659 | 0.663 | 0.659 | 0.655 |
| | LtCMH (AAAI'23) | 0.706 | 0.718 | 0.725 | 0.731 | 0.688 | 0.695 | 0.718 | 0.724 | 0.671 | 0.708 | 0.714 | 0.722 |
| | **Our NRCH** | **0.748** | **0.762** | **0.764** | **0.768** | **0.740** | **0.754** | **0.762** | **0.766** | **0.731** | **0.738** | **0.754** | **0.754** |

(a) MIRFlickr-25K (I2T)  (b) MS-COCO (I2T)  (c) NUS-WIDE (I2T)  (d) IAPR TC-12 (I2T)

(e) MIRFlickr-25K (T2I)  (f) MS-COCO (T2I)  (g) NUS-WIDE (T2I)  (h) IAPR TC-12 (T2I)

**Figure 4: The precision-recall curves on four datasets. Note that the length of hash codes is 64 and the noise rate is 50%.**

Deriving insights from the experimental result in Tables 1 and 2, the subsequent observations can be formulated:

- As the noise rate increases, the performance of these supervised methods [7, 15, 31, 35, 37, 41–43] degrades. In comparison, the unsupervised methods [11, 27, 38] on the dotted line in the tables seem to have a certain degree of robustness. However, it is still difficult for them to achieve further performance improvement as they do not use labels for training.

- Among all these baseline methods, CMMQ [37] stands out for its resistance to noisy labels on the NUS-WIDE and MS-COCO datasets, sharing similarities with our NRCH in terms of noise segregation components. But unlike this, our NRCH achieves even more promising performance by improving the robustness of the loss and performing reliable dynamic sample selection.

- All in all, our NRCH surpasses all baselines on four datasets and outperforms the best baselines by 2.9%, 4.5%, 6.2%, and

0.8%, respectively, in the most challenging scenarios (*i.e.*, the noise rate is 80% noise and code length is 16 bits). This is enough to prove the effectiveness and superiority of our NRCH against noisy labels.

## 4.5 Ablation Study

In this section, we perform an ablation study on the MIRFlickr-25K dataset with a noise rate of 50% to assess the effectiveness of the proposed components (DNS and $R(\mathcal{D}_n)$) in cross-modal retrieval. For a comprehensive exploration of the contribution of each component, we contrast our NRCH with its three variants: (1) NRCH devoid of any components, (2) NRCH with only $R(\mathcal{D}_n)$, and (3) NRCH with solely DNS. To ensure an equitable comparison, all comparative variants are trained using the same configurations as our NRCH. To further affirm the model's capacity to endure noise interference, we selected outcomes from the final epoch of training and evaluated them on the test dataset. As illustrated in Table 3, the full version of NRCH (4) shows the best performance while other variants have suboptimal results, which means that all proposed components are vital for NRCH.

**Table 3: Ablation studies on the MIRFlickr-25K dataset with 50% noise. The highest and the second-highest scores are in bold and underlined, respectively.**

| Configuration | | Image to Text | | | | Text To Image | | | |
|---|---|---|---|---|---|---|---|---|---|
| No. DNS | $R(\mathcal{D}_n)$ | 16bit | 32bit | 64bit | 128bit | 16bit | 32bit | 64bit | 128bit |
| (1) X | X | 0.739 | 0.745 | 0.755 | 0.757 | 0.724 | 0.732 | 0.749 | 0.747 |
| (2) X | ✓ | 0.742 | 0.751 | 0.758 | 0.762 | 0.725 | 0.737 | 0.752 | 0.749 |
| (3) ✓ | X | 0.744 | 0.754 | 0.759 | 0.765 | 0.729 | 0.741 | 0.753 | 0.754 |
| (4) ✓ | ✓ | **0.747** | **0.762** | **0.770** | **0.772** | **0.734** | **0.747** | **0.754** | **0.759** |

## 4.6 Robustness Study

To study the robustness of our NRCH intuitively, we select the baseline CMMQ and a variant of NRCH without DNS as the comparative approaches. Then, we splot MAP scores versus epochs on the test dataset in Figure 5. The results indicate that while CMMQ may enhance their initial training performance, their susceptibility to noisy label interference results in a lower MAP compared to our NRCH. Meanwhile, the absence of a dynamic strategy to identify noisy labels leads CMMQ to a serious overfitting issue in the late stages of training, which would precipitate significant performance deterioration as shown in Figure 5. On the contrary, our NRCH manifests a continual enhancement in efficacy throughout the initial training phases and upholds steadiness without a noteworthy downturn in the subsequent stages. The variant also suffers a performance drop due to its inability to discern and retain confident samples, thus leading to suboptimal MAP scores. Overall, compared with CMMQ and the variant, our proposed method has an effective dynamic separation strategy and can achieve much superior robustness and better results.

## 4.7 Parameter Sensitivity Analysis

In this subsection, we investigate the sensitivity of the parameter $\lambda$ on the MIRFlickr-25k and IAPR TC-12 datasets. We report the results with a hash code length of 64 bits under 50% noise rate and $\lambda$ is adjusted in the range of 0.1 to 1. The outcomes are illustrated in Figure 6, where "Best" represents the best average MAP value on

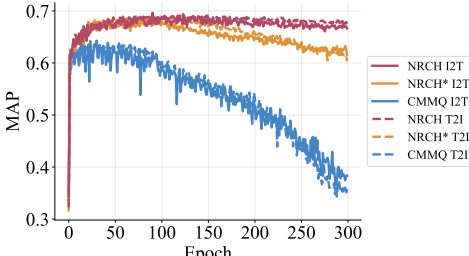

**Figure 5: Test MAP scores versus epochs on the MS-COCO dataset with 50% noise. Note that the length of hash codes is 64 and NRCH\* means the variant of NRCH without DNS.**

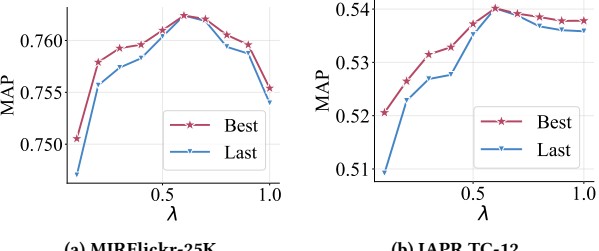

(a) MIRFlickr-25K       (b) IAPR TC-12

**Figure 6: Sensitive analysis on the MIRFlickr-25K and IAPR TC-12 datasets with 50% noise. The hash code length is 64.**

both I2T and T2I tasks, and "Last" means the average MAP value for the last training epoch. As depicted in Figure 6, the trend for both peak and long-term training results starts with an upward trajectory, stabilizing at higher levels once $\lambda$ exceeds 0.6, before eventually tapering off. Interestingly, the curves represented by the MAP results of the best and final training sessions almost overlap each other when the value of $\lambda$ is between 0.6 and 0.7. This suggests that our model demonstrates strong robustness when $\lambda$ is within the range of 0.6 to 0.7, effectively mitigating the problem of deep network overfitting to noisy labels. However, as the value of $\lambda$ deviates from this range, the gap between the two curves increases, indicating a decrease in robustness. Furthermore, configuring $\lambda$ as 1 leads to the deterioration of our model into a setup devoid of the binary regularization term, which substantiates the significance of the binary regularization term in our method. Thus, opting for $\lambda$ values within the range of 0.6 to 0.7 is advisable. In our experiments, we designate $\lambda$ as 0.6.

## 5 Conclusion

In this paper, we propose a novel cross-modal hashing approach to learn from noisy labels, *i.e.*, NRCH. NRCH is equipped with two parts, *i.e.*, the Robust Contrastive Hashing loss (RCH) and the Dynamic Noise Separator (DNS). RCH directs our model to focus on more reliable positive pairs instead of noisy ones, thus avoiding overfitting to noisy labels. DNS enables us to separate the clean and noisy samples dynamically and train our model with more reliable samples, thus embracing robustness against noisy labels. Diverging from previous methods, our NRCH can dynamically discriminate the clean and corrupted labels with loss distributions per sample while alleviating the error accumulation. Extensive experiments conducted on four widely used benchmarks reveal that our NRCH outperforms existing state-of-the-art methods under noisy labels and shows strong robustness.

## Acknowledgments

This work was supported by NSFC under Grants U21B2040, 62176171, 62102274, 62372315; by Sichuan Science and Technology Planning Project under Grants 2024NSFTD0038, 2024NSFTD0047, 24ZDZX0007, 2024NSFTD0049, 2023ZYD0143, 2024YFHZ0144, 2024YFHZ0089; by the Fundamental Research Funds for the Central Universities under Grants CJ202303 and CJ202403.

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
