# OpenReview forum: "Robust Contrastive Cross-modal Hashing with Noisy Labels"
_acmmm.org/ACMMM/2024/Conference — MM2024 Poster_

### Official Review · Reviewer_DpGt · 2024-05-09

**Rating:** 3
**Confidence:** 3

**Summary:**

This paper designs a Noise Resistance Cross-modal Hashing to learn hashing with noisy labels. First, the author propose a Robust Contrastive Hashing loss to focus on homologous pairs rather than noisy positive ones. Second, a novel Dynamic Noise Separator is introduced to dynamically discriminate the clean and noisy labels based on the loss distributions.

**Strengths:**

1. The presentation of this motivation is clear and easy to follow.
2. The experiments are adequate and can sufficiently demonstrate the effectiveness of NRCH, especially considering that the author conducted validations on a plethora of baselines (11 in total) across five datasets.

**Limitations:**

1. In Section 3.2, I don’t understand the underlying rationale for transitioning from Equation 2 to Equation 6. In Equation (4), the author establishes an upper bound, hence the objective is to optimize this bound. However, wouldn't directly optimizing Equation (2) be more straightforward?
2. The novelty is constrained. While the author may be among the first to apply the memorization effect to cross-model hashing, this method is quite prevalent in the domain of label noise. For instance, "Robust Learning by Self-Transition for Handling Noisy Labels" delves further into the study of the memorization effect. The author should elucidate the distinct impact of this effect on binary representations like hashing compared to real-value representations.
3. In the experiment, the authors maintained the backbones frozen throughout the training phase. Nevertheless, a significant aspect of the impact of label noise on the model lies in the issue of model parameters memorizing such noise. Would the act of freezing have an effect on this particular characteristic?
4. In Equation 1, the author employs the sign function to obtain the hash code, yet it appears that the issue of differentiation is not explicitly addressed. While methods to address this concern in hashing are commonplace, providing clear elucidation would be advantageous for the reproducibility of the approach.

**Suitability:**

3

---

### Official Review · Reviewer_y4kK · 2024-05-17

**Rating:** 5
**Confidence:** 4

**Summary:**

This paper proposed a new robust framework for solving the noisy label problem in cross-modal hashing, which contains a novel Robust Contrastive Hashing loss and a  Dynamic Noise Separator for practicing robust cross-modal hashing, named Noise Resistance Cross-modal Hashing. Judging from the resulting performance, the proposed method seems to be effective. Overall, this paper is well-organized and well-experimented. But I have some questions about this paper that need to be clarified or solved by the authors, see Limitations.

**Strengths:**

1. This paper studied an important issue in cross-modal hash learning, namely noisy labels.

2. The experiments in this paper are sufficient.

3. The ablation experiment appears to be effective.

4. The details of the experiment are disclosed and I think it can be reproduced.

**Limitations:**

1. Like some works on noisy labels, I think the authors should report multiple average results 1 (mean+std) or add and report the test results of the last epoch. This can avoid random effects.

2. Lack of ablation exploration of warmup (Eq.16).

3. The author should provide more visual results on all datastes like  Fig. 5 to prove the generalization and robustness of the proposed methods.

4. Is the learnable parameter matrix mentioned in line 415 randomly initialized? Does different initialization have any impact on final performance? For example, orthogonal initialization.

5. Is there a baseline proposed in 2024? It would be better if there is. (Does not affect my rating)

**Suitability:**

3

---

### Official Review · Reviewer_W4fk · 2024-05-20

**Rating:** 5
**Confidence:** 3

**Summary:**

This paper proposes a NRCH approach for cross-modal retrieval to learn from noisy labels, which has two parts, i.e., the Robust Contrastive Hashing loss (RCH) and the Dynamic Noise Separator (DNS). Specifically, RCH directs the model to focus on more reliable positive pairs instead of noisy ones, thus avoiding overfitting to noisy labels. DNS enables us to separate the clean and noisy samples dynamically and train the model with more reliable samples, thus embracing robustness against noisy labels. Extensive experiments shows the proposed NRCH outperforms existing state-of-the-art methods.

**Strengths:**

1.The idea is interesting and the proposed method seems to be novel.
2.This paper has clear motivations and technical solutions.
3.The proposed method achieves good results.

**Limitations:**

1.The proposed loss is more robust than triplet loss, which is not explained in the paper.
2.The article says that unsupervised methods lack corresponding measures for noisy labels. This description is problematic.
3.The article only provides the ablation experiment results on one dataset. Can you provide the ablation experiment results on other datasets?
4.The paper proposes a binary regularization term, but in the parameter sensitivity experimental analysis, when \lambda is 1 (no regularization term) on the IAPR TC-12 dataset, the performance does not change much. Can you give the results of other datasets without regularization terms?
5.It is recommended to add some experiments in the ablation experiment. Replace the proposed loss with other losses, such as CE, MAE, triplet loss.

**Suitability:**

3

---

### Official Review · Reviewer_mBfG · 2024-05-24

**Rating:** 3
**Confidence:** 4

**Summary:**

This paper proposes the Noise Resistance Cross-modal Hashing (NRCH) to implement hashing learning with noisy labels. The core contribution is to overcome two key challenges, i.e., noise overfitting and error accumulation.

**Strengths:**

The paper is well-written and easy to follow, with detailed quantitative and qualitative experimental results.

**Limitations:**

1. This paper utilizes a series of supervised and unsupervised methods for comparison; however, it does not consider or compare related work on hashing under label noise. For instance, [1, 2] have both explored contrastive learning in Hamming space.

2. There is a lack of experiments on different types of noise. [1, 2] consider both pairflip and symmetric noise for a comprehensive study.

[1] Sun J, Wang H, Luo X, et al. Heart: Towards effective hash codes under label noise[C]//Proceedings of the 30th ACM International Conference on Multimedia. 2022: 366-375.
[2] Wang H, Jiang H, Sun J, et al. DIOR: Learning to Hash With Label Noise Via Dual Partition and Contrastive Learning[J]. IEEE Transactions on Knowledge and Data Engineering, 2023.

**Suitability:**

3

---

### Meta-Review · Area_Chair_DR2h · 2024-06-27

**Recommendation:** Accept (Poster)
**Confidence:** 4

**Metareview:**

According to all the review comments, rebuttals, discussions and final ratings, the majority of the reviewers gave positive ratings to this paper and the concerns were well addressed. I am happy to recommend to accept this paper. Please carefully revise the final manuscript according to the comments and discussions.